# EMFlow: Data Imputation in Latent Space via EM and Deep Flow Models

## Abstract

The presence of missing values within high-dimensional data is an ubiquitous problem for many applied sciences. A serious limitation of many available data mining and machine learning methods is their inability to handle partially missing values and so an integrated approach that combines imputation and model estimation is vital for down-stream analysis. A computationally fast algorithm, called EMFlow, is introduced that performs imputation in a latent space via an online version of Expectation-Maximization (EM) algorithm by using a normalizing flow (NF) model which maps the data space to a latent space. The proposed EMFlow algorithm is iterative, involving updating the parameters of online EM and NF alternatively. Extensive experimental results for high-dimensional multivariate and image datasets are presented to illustrate the superior performance of the EMFlow compared to a couple of recently available methods in terms of both predictive accuracy and speed of algorithmic convergence.

## 1 Introduction

Missing values are often encountered for real-world datasets and are known to adversely impact the validity of down-stream analysis. Most machine learning (ML) algorithms and statistical tools often ignore the problem of partially observed features by simply dropping entire cases with missing values, which could result into biased estimation and underestimation of parameter uncertainty (Schmitt et al., 2015; Somasundaram & Nedunchezhian, 2011; Lall, 2016).

There have been some recent attempts to develop ML methods that properly handle missing values. Early works in this field performed imputation in a supervised manner where a complete training set is needed to learning the correlation between missing and observed entries (e.g. García-Laencina et al., 2010; Rezende et al., 2014; Bertalmio et al., 2000; Xie et al., 2012; Yeh et al., 2016). However, it is common that a large collection of fully observed data is hard to acquire, which makes those data-greedy supervised algorithms less effective. On the other hand, early attempts in unsupervised imputation methods, such as the collaborative filtering (Sarwar et al., 2001), usually learn the correlation across dimensions by embedding the data into a lower dimensional latent space linearly (e.g. Little & Rubin, 2019; Audigier et al., 2016). To improve the limited representation power of those linear models, kernel-based methods are also proposed but at the cost of excessive computational load when dealing with large datasets (e.g. Sanguinetti & Lawrence, 2006; Liu et al., 2016).

More recently, several imputation methods based on deep generative models have been proposed, providing much more accurate estimates of missing values especially for high-dimensional image datasets. In this work, we introduce *EMFlow* that integrates the normalizing flow (NF) (Dinh et al., 2016; Rezende & Mohamed, 2015; Dinh et al., 2014; Kingma & Dhariwal, 2018) with an online version of Expectation-Maximization (EM) algorithm (Cappé & Moulines, 2009). The proposed framework is motivated by the strength and weakness of EM. As a class of iterative algorithm designed for latent variable models, EM can be applied to data imputation in an interpretable way (Little & Rubin, 2019). Additionally, the learning of EM is usually numerically stable and its convergence property has been studied extensively (e.g. Ma et al., 2000; Zhao et al., 2020; Meng et al., 1994; Wu, 1983). However, the E-step and M-step are only traceable with simple underlying distributions including multivariate Gaussian or Student's $t$ distributions as well as their mixtures (e.g. Di Zio et al., 2007; xian Wang et al., 2004). On the other hand, NF is capable of latent representation and efficient data sampling, which makes it a convenient bridge between the data space and the

latent space. Therefore, we let EM perform imputation in the latent space where simple inter-feature dependency is assumed (i.e. the underlying distribution is multivariate Gaussian). Meanwhile, NF is used to recover the mapping between the complex inter-feature dependency in the data space and the one in the latent space learned by EM.

The inference of EMFlow adopts an iterative learning strategy that has been widely used in model-based multiple imputation methods where the initial naive imputation is refined step by step until convergence (e.g. Gondara & Wang, 2017; Buuren & Groothuis-Oudshoorn, 2010; Stekhoven & Bühlmann, 2012). Specifically, three steps are performed alternatively:

- update the density estimation of complete data including the observed and current imputed values;
- update the base distribution (i.e. $\mu$ and $\Sigma$) in the latent space by EM; and
- update the imputation in the data space.

Note that the first update corresponds to optimizing the complete data likelihood as well as the reconstruction error computed on the observed entries. We also derive an online version of EM such that it only consumes a batch of data at a time for parameter updates and thus can work with deep generative models smoothly. We will show that such learning schema has simpler implementation and leads to faster convergence compared to other competing methods.

The main contributions of this work are

(i) an imputation framework combining an online version of Expectation-Maximization (EM) algorithm and the normalizing flow;
(ii) an iterative learning schema that alternatively updates the inter-feature dependency in the latent space (i.e. the base distribution) and density estimation in the data space;
(iii) a derivation of online EM in the context of missing data imputation; and
(iv) extensive experiments on multiple image and tabular datasets to demonstrate the imputation quality and convergence speed of the proposed framework.

## 2 RELATED WORK

Recently, the applications of deep generative models like Generative Adversarial Networks (GAN) (Goodfellow et al., 2014) have been extended to the field of missing data imputation under the assumption of Missing at Random (MAR) or Missing Completely at Random (MCAR). GAIN (Yoon et al., 2018) designs a compete data generator that performs imputation, and a discriminator to differentiate imputed and observed components with the help of a *hint* mechanism. MisGAN (Li et al., 2019) introduces another pair of generator-discriminator that aims to learn the distribution of missing mask. However, training GAN-based models is a notoriously challenging for its excessively complex structure and non-convex objectives. For example, MisGAN optimizes three objectives jointly involving six neural networks for imputation. Furthermore, GAN-based models do not obtain explicit density estimation that could be critical for down-stream analysis (e.g. Ferdosi et al., 2011; Zambom & Ronaldo, 2013).

Some imputation techniques based on Variational Autoencoders (VAE) (Kingma & Welling, 2013) are also developed. For example, MIWAE Mattei & Frellsen (2019) leverages the importance-weighted autoencoder (Burda et al., 2015) and optimizes a lower bound of the likelihood of observed data. But the zero imputation adopted by it can be problematic since observed entries can also be zero. It also needs quite large computational power to make the bound to be tight. EDDI (Ma et al., 2018) avoids zero imputation by introducing a permutation invariant encoder. Imputation under Missing not at Random (MNAR) has also been explored by explicitly modeling the missing mechanism (Ipsen et al., 2020; Collier et al., 2020). However, all VAE-based approaches only permit approximate density estimation no matter how expressive the inference model is.

Our proposed framework builds on the work of MCFlow (Richardson et al., 2020) that utilizes NF to learn exact density estimation on incomplete data via an iterative learning schema. The core component of MCFlow is a feed forward network that operates in the latent space and attempts to find the likeliest embedding vector in a supervised manner. Although MCFlow achieves impressive performance compared to other state-of-art methods, it remains ambiguous that how the feed forward

network exploits the correlation in the latent space. A key difference between the MCFlow and our proposed method is that we exploit the correlation in the latent space which in turn induces correlation in ambient space via NF. We also find that the inference of MCFlow often has slow convergence speed and can be unstable.

## 3 APPROACH

### 3.1 PROBLEM DEFINITION

We define the complete dataset $\mathbf{X}$ as a collection of $p$-dimensional vectors $\{\mathbf{x}_1, \ldots, \mathbf{x_n}\}$ that are independent and identically distributed (i.i.d.) samples drawn from $p_X(\cdot; \theta)$ in a $p$-dimensional data space $\mathcal{X}$. The missing pattern of $\mathbf{x}_i$ is described by a binary mask $\mathbf{m}_i \in \{0, 1\}^p$ such that $\mathbf{x}_{ij}$ is missing if $\mathbf{m}_{ij} = 1$, and $\mathbf{x}_{ij}$ is observed if $\mathbf{m}_{ij} = 0$.

Let $\mathbf{x}^o$ and $\mathbf{x}^m$ be the observed and missing parts of $\mathbf{x}$, and $p_M(\mathbf{m}|\mathbf{x}) = p_M(\mathbf{m}|\mathbf{x}^o, \mathbf{x}^m)$ be the conditional distribution of the mask. Based on dependency between $\mathbf{m}$ and $(\mathbf{x}^o, \mathbf{x}^m)$, the missing mechanism can be classified into three classes (Little & Rubin, 2019):

- MCAR: $p_M(\mathbf{m}|\mathbf{x}^o, \mathbf{x}^m) = p_M(\mathbf{m})$
- MAR: $p_M(\mathbf{m}|\mathbf{x}^o, \mathbf{x}^m) = p_M(\mathbf{m}|\mathbf{x}^o)$
- MNAR: The probability of missing depends on both $\mathbf{x}^o$ and $\mathbf{x}^m$.

Throughout this paper, we only focus on MCAR or MAR where the missing mechanism can be safely ignored. The typical objective of learning a latent model is to maximize the observed data log-likelihood defined as

$$L^{obs}(\theta) = \sum_{i=1}^n \log \int p_X(\mathbf{x}_i^o, \mathbf{x}_i^m; \theta) \, d\mathbf{x}_i^m. \tag{1}$$

However, our goal in this work is to estimate $p_X$ from incomplete data as well as obtain accurate imputation under $p_X$. Therefore, we attempt to learn the reconstructed data $\widehat{\mathbf{X}} = (\widehat{\mathbf{x}}_1, \ldots, \widehat{\mathbf{x}}_n)^T$ and the estimated model parameter $\widehat{\theta}$ via

$$(\widehat{\mathbf{X}}, \widehat{\theta}) = \arg\max_{\mathbf{x}_i \in \mathcal{X}_i', \theta} \sum_{i=1}^n \log p_X(\mathbf{x}_i | \theta) \tag{2}$$

where $\mathcal{X}_i' \subseteq \mathcal{X}$ is the search space for $\mathbf{x}_i$ where the observed locations have fixed values.

### 3.2 NORMALIZING FLOWS

To make it possible to optimize equation 2, $p_X(\mathbf{x}; \theta)$ needs to be specified in a parametric way that should be expressive enough, as the density is potentially complex and high-dimensional. To this end, we use NF to model $p_X(\mathbf{x}; \theta)$ as an invertible transformation $f_\psi$ of a base distribution $p_Z(\mathbf{z}; \phi)$ in the latent space $\mathcal{Z}$. Under the change of variables theorem, the complete data log-density is specified as

$$p_X(\mathbf{x}; \theta) = p_Z(f_\psi^{-1}(\mathbf{x}); \phi) \left| \det\left( \frac{\partial f_\psi^{-1}(\mathbf{x})}{\partial \mathbf{x}^T} \right) \right| \tag{3}$$

where $\theta = (\psi, \phi)$.

$f_\psi$ is usually composed by a sequence of relatively simple transformations to approximate arbitrarily complex distributions with high representation power. In this work, we choose Real NVP (Dinh et al., 2016) based on affine coupling transformations as the flow model[1]. Recently, Teshima

---

[1]See appendix A for the details of Real NVP.

et al. (2020) has shown that flow models constructed from affine coupling layers can be universal distributional approximators.

Data imputation relies on the inter-feature dependency that is usually intractable and hard to capture in the data space $\mathcal{X}$ with the presence of missing entries. Therefore, we make the following two assumptions related to NF.

**Assumption 1:** The inter-feature dependency in the latent space $\mathcal{Z}$ is simple and can be characterized by a multivariate Gaussian density, that is:

$$p_Z(\mathbf{z}; \phi) = \mathcal{N}(\mathbf{z}; \boldsymbol{\mu}, \boldsymbol{\Sigma}) \tag{4}$$

where $\phi = (\boldsymbol{\mu}, \boldsymbol{\Sigma})$ consists of latent parameters; mean vector $\boldsymbol{\mu}$ and covariance matrix $\boldsymbol{\Sigma}$.

Note that the base distribution is usually chosen as a standard Gaussian distribution (with $\boldsymbol{\mu} = \mathbf{0}$ and $\boldsymbol{\Sigma} = \boldsymbol{I}_p$) in the literature for simplicity. However, the covariance $\boldsymbol{\Sigma}$ is essential in our imputation work to represent the inter-feature dependency.

**Assumption 2:** Given that the transformations involved in NF are feature-wise (e.g. Dinh et al., 2016; 2014), we expect NF to learn the mapping between the complex inter-feature dependency in the data space $\mathcal{X}$ and the simple one in the latent space $\mathcal{Z}$.

### 3.3 ONLINE EM

EM is a class of iterative algorithms for latent variable models including missing data imputation(Little & Rubin, 2019). In EMFlow, it works in the latent space $\mathcal{Z}$ where the underlying distribution is $\mathcal{N}(\mathbf{z}; \boldsymbol{\mu}, \boldsymbol{\Sigma})$. Given the embedding vectors $\{\mathbf{z}_1, \ldots, \mathbf{z}_n\}$ where $\mathbf{z}_i = f_\psi^{-1}(\mathbf{x}_i)$, and the corresponding missing mask $\{\mathbf{m}_1, \ldots, \mathbf{m}_n\}$, EM aims to estimate $(\boldsymbol{\mu}, \boldsymbol{\Sigma})$ in an iterative way:

$$\begin{aligned}
\widehat{\boldsymbol{\mu}}^{(t+1)} &= g_{\boldsymbol{\mu}}(\widehat{\boldsymbol{\mu}}^{(t)}, \widehat{\boldsymbol{\Sigma}}^{(t)}; \{\mathbf{z}_i, \mathbf{m}_i\}_{i=1}^n) \\
\widehat{\boldsymbol{\Sigma}}^{(t+1)} &= g_{\boldsymbol{\Sigma}}(\widehat{\boldsymbol{\mu}}^{(t)}, \widehat{\boldsymbol{\Sigma}}^{(t)}; \{\mathbf{z}_i, \mathbf{m}_i\}_{i=1}^n)
\end{aligned} \tag{5}$$

where $(\widehat{\boldsymbol{\mu}}^{(t)}, \widehat{\boldsymbol{\Sigma}}^{(t)})$ are the estimates at the $t^{th}$ iteration, and $\{g_{\boldsymbol{\mu}}(\cdot), g_{\boldsymbol{\Sigma}}(\cdot)\}$ denote the mappings between two consecutive iterations [2].

Given estimates $(\widehat{\boldsymbol{\mu}}, \widehat{\boldsymbol{\Sigma}})$, the missing part $\mathbf{z}_i^m$ is imputed by its conditional mean given the observed part $\mathbf{z}_i^o$ :

$$\widehat{\mathbf{z}}_i^m = E(\mathbf{z}_i^m | \mathbf{z}_i^o; \widehat{\boldsymbol{\mu}}, \widehat{\boldsymbol{\Sigma}}) = \widehat{\boldsymbol{\mu}}_{\mathbf{m}_i} + \widehat{\boldsymbol{\Sigma}}_{\mathbf{m}_i \mathbf{o}_i} \left( \widehat{\boldsymbol{\Sigma}}_{\mathbf{o}_i \mathbf{o}_i} \right)^{-1} (\mathbf{z}_i^o - \widehat{\boldsymbol{\mu}}_{\mathbf{o}_i}) \tag{6}$$

where $\mathbf{o}_i$ is the observed mask (i.e. the complement of $\mathbf{m}_i$), and the subscripts of $(\widehat{\boldsymbol{\mu}}, \widehat{\boldsymbol{\Sigma}})$ denote the slicing indexes.

When processing datasets of large volume, EM becomes impractical because it needs to read the whole data into the memory for each iteration. Following the framework introduced by Cappé & Moulines (2009), we derive an online version of EM algorithm in the context of data imputation. Let $B \subset \{1, \ldots, n\}$ denote a mini-batch of sample indexes, the online EM first obtains *local* estimates from a batch of data:

$$\begin{aligned}
\widehat{\boldsymbol{\mu}}_{local} &= g_{\boldsymbol{\mu}}(\widehat{\boldsymbol{\mu}}^{(t)}, \widehat{\boldsymbol{\Sigma}}^{(t)}; \{\mathbf{z}_i, \mathbf{m}_i\}_{i \in B}) \\
\widehat{\boldsymbol{\Sigma}}_{local} &= g_{\boldsymbol{\Sigma}}(\widehat{\boldsymbol{\mu}}^{(t)}, \widehat{\boldsymbol{\Sigma}}^{(t)}; \{\mathbf{z}_i, \mathbf{m}_i\}_{i \in B})
\end{aligned} \tag{7}$$

The *global* estimates are then updated in the fashion of weighted average:

$$\begin{aligned}
\widehat{\boldsymbol{\mu}}^{(t+1)} &= \rho_{t+1} \widehat{\boldsymbol{\mu}}_{local} + (1 - \rho_{t+1}) \widehat{\boldsymbol{\mu}}^{(t)} \\
\widehat{\boldsymbol{\Sigma}}^{(t+1)} &= \rho_{t+1} \widehat{\boldsymbol{\Sigma}}_{local} + (1 - \rho_{t+1}) \widehat{\boldsymbol{\Sigma}}^{(t)}
\end{aligned} \tag{8}$$

---

[2]See appendix B for the details of $g_{\boldsymbol{\mu}}$ and $g_{\boldsymbol{\Sigma}}$.

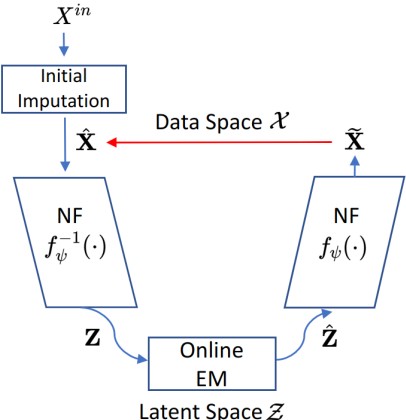

Figure 1: EMFlow architecture.

where $(\rho_1, \rho_2, \ldots)$ are a sequence of step sizes satisfying

$$0 < \rho_t < 1, \sum_{i=1}^{\infty} \rho_i = \infty \text{ and } \sum_{i=1}^{\infty} \rho_i^2 < \infty \tag{9}$$

In this work, we use a step size schedule defined by

$$\rho_t = Ct^{-\gamma}, \quad t = 1, 2, \ldots \tag{10}$$

where $C$ is a positive constant and $\gamma \in (0.5, 1]$.

### 3.4 ARCHITECTURE AND INFERENCE

EMFlow is a composite framework that combines NF and online EM. As illustrated in Fig 1, NF is the bidirectional tunnel between the data space and the latent space, aiming to learn the complete data density $p_X$. In the latent space, the online EM estimates the inter-feature dependency of the embedding vectors and performs imputation. To address the issue that NF needs complete data vectors for computation, the incomplete data $\mathbf{X}^{in}$ are imputed naively (e.g. median imputation for tabular datasets) at the very beginning to get the *initial* current imputed data $\widehat{\mathbf{X}}$. Afterwards, the objective in equation 2 is optimized in an iterative schema, where each iteration consists of a training phase and a re-imputation phase.

**Training Phase** At this phase, the current imputed data $\widehat{\mathbf{X}}$ stay fixed, while the parameter estimates of NF (i.e. $\widehat{\psi}$) and base distribution (i.e. $\widehat{\boldsymbol{\mu}}$ and $\widehat{\boldsymbol{\Sigma}}$) are updated in different ways.

First of all, given the current estimated base distribution $\mathcal{N}(\cdot; \widehat{\mu}, \widehat{\Sigma})$, the flow model $f_\psi$ are learned by minimizing the negative log-likelihood of a batch of the current imputed data $\widehat{\mathbf{X}}_B$:

$$L_1(\psi) = -\frac{1}{|B|} \sum_{i \in B} \log p_X(\widehat{\mathbf{x}}_i; \psi, \widehat{\boldsymbol{\mu}}, \widehat{\boldsymbol{\Sigma}}) \tag{11}$$

where $|B|$ denotes the batch size.

The computation of $L_1$ requires exact likelihood evaluation that is equipped by NF. Once the flow model parameters get updated, we obtain the embedding vectors in the latent space:

$$\mathbf{z}_i = f_{\widehat{\psi}}^{-1}(\widehat{\mathbf{x}}_i), \quad i \in B \tag{12}$$

Although the embedding vectors are complete, they are treated as incomplete using the missing masks in the data space $\{\mathbf{m}_i\}_{i \in B}$, given that the invertible mapping parameterized by $f_\psi$ is feature-wise. Therefore, the online EM imputes the missing parts of the embedding vectors with the current global estimates $(\widehat{\boldsymbol{\mu}}, \widehat{\boldsymbol{\Sigma}})$:

$$\widehat{\mathbf{z}}_i^m = E(\mathbf{z}_i^m | \mathbf{z}_i^o; \widehat{\boldsymbol{\mu}}, \widehat{\boldsymbol{\Sigma}}), \quad i \in B \tag{13}$$

which results in new embedding vectors $\{\widehat{\mathbf{z}}\}_{i \in B}$ where $\widehat{\mathbf{z}}_i$ consists of the observed part $\mathbf{z}_i^o$ and the imputed part $\widehat{\mathbf{z}}_i^m$. After the imputation, the global estimates $(\widehat{\boldsymbol{\mu}}, \widehat{\boldsymbol{\Sigma}})$ are also updated following equation 7 and equation 8.

Since the base distribution has been changed, it's necessary to update the flow model $f_\psi$ again by optimizing a composite loss:

$$L_2(\psi) = -\frac{1}{|B|} \sum_{i \in B} \left[ \log p_X(\widetilde{\mathbf{x}}_i; \psi, \widehat{\boldsymbol{\mu}}, \widehat{\boldsymbol{\Sigma}}) - \alpha L_{\text{rec}}(\widetilde{\mathbf{x}}_i, \widehat{\mathbf{x}}_i, \mathbf{m}_i) \right] \tag{14}$$

where $\widetilde{\mathbf{x}}_i = f_\psi(\widehat{\mathbf{z}}_i)$, and $L_{\text{rec}}(\widetilde{\mathbf{x}}_i, \widehat{\mathbf{x}}_i, \mathbf{m}_i)$ is the reconstruction error only for non-missing values:

$$L_{\text{rec}}(\widetilde{\mathbf{x}}_i, \widehat{\mathbf{x}}_i, \mathbf{m}_i) = \sum_{j=1}^{p} (1 - \mathbf{m}_{ij})(\widetilde{\mathbf{x}}_{ij} - \widehat{\mathbf{x}}_{ij})^2 \tag{15}$$

In this composite loss, the first term forces the reconstructed data vectors $\{\widetilde{\mathbf{x}}\}_{i \in B}$ to have high likelihood in the data space, while the second term encourages $\{\widetilde{\mathbf{x}}\}_{i \in B}$ to match the observed parts of the incomplete training data. And since $\{\widetilde{\mathbf{x}}\}_{i \in B}$ are transformed from $\{\widehat{\mathbf{z}}\}_{i \in B}$ via $f_\psi$, both terms are conditioned on the inter-feature dependency learned by EM. A pseudocode for this training phase is presented in Algorithm 1, and the implementation details that aim to make the training phase more stable is presented in appendix D.

---

**Algorithm 1** Training Phase

---

1: **Input:** Current imputation: $\widehat{\mathbf{X}} = (\widehat{\mathbf{x}}_1, \ldots, \widehat{\mathbf{x}}_n)^T$, missing masks $\mathbf{M} = (\mathbf{m}_1, \ldots, \mathbf{m}_n)^T$, initial estimates of the base distribution $(\widehat{\boldsymbol{\mu}}^{(0)}, \widehat{\boldsymbol{\Sigma}}^{(0)})$, online EM step size sequence: $\rho_1, \rho_2, \ldots, \rho_t, \ldots$
2: **for** $t = 1$ to $T_{\text{epoch}}$ **do**
3:     Get a mini-batch $\widehat{\mathbf{X}}_B = \{\widehat{\mathbf{x}}_i\}_{i \in B}$
4:     # update the flow model
5:     Compute $L_1$ in equation 11
6:     Update $\psi$ via gradient descent
7:     # update the base distribution
8:     $\mathbf{z}_i = f_\psi^{-1}(\widehat{\mathbf{x}}_i), \quad i \in B$
9:     Impute in the latent space with $(\widehat{\boldsymbol{\mu}}^{(t-1)}, \widehat{\boldsymbol{\Sigma}}^{(t-1)})$ to get $\{\widehat{\mathbf{z}}_i\}_{i \in B}$ via equation 13
10:    Obtain updated $(\widehat{\boldsymbol{\mu}}^{(t)}, \widehat{\boldsymbol{\Sigma}}^{(t)})$ via equation 7 and equation 8
11:    # update the flow model again
12:    $\widetilde{\mathbf{x}}_i = f_\psi(\widehat{\mathbf{z}}_i), \quad i \in B$
13:    Compute $L_2$ via equation 14
14:    Update $\psi$ via gradient descent

---

**Re-Imputation Phase** After the training phase, the re-imputation phase is executed to update the current imputation $\widehat{\mathbf{X}}$. This procedure is similar to that in the training phase, except that all model parameters are kept fixed. As shown by the last line of Algorithm 2, the missing parts of $\widehat{\mathbf{X}}$ are replaced with those of the reconstructed data vectors, while the observed parts are kept the same.

---

**Algorithm 2** Re-imputation Phase

---

1: **Input:** Current imputation: $\widehat{\mathbf{X}} = (\widehat{\mathbf{x}}_1, \ldots, \widehat{\mathbf{x}}_n)^T$, missing masks $\mathbf{M} = (\mathbf{m}_1, \ldots, \mathbf{m}_n)^T$, estimates of the base distribution $(\widehat{\boldsymbol{\mu}}, \widehat{\boldsymbol{\Sigma}})$ from the the previous training phase
2: **for** $i = 1$ to $n$ **do**
3:      $\mathbf{z}_i = f_\psi^{-1}(\widehat{\mathbf{x}}_i)$
4:      Impute in the latent space with $(\widehat{\boldsymbol{\mu}}, \widehat{\boldsymbol{\Sigma}})$ to get $\widehat{\mathbf{z}}_i$ via equation 13
5:      $\widetilde{\mathbf{x}}_i = f_\psi(\widehat{\mathbf{z}}_i)$
6:      # update current imputation
7:      $\widehat{\mathbf{x}}_i = \widehat{\mathbf{x}}_i \odot \mathbf{o}_i + \widetilde{\mathbf{x}}_i \odot \mathbf{m}_i$

---

Table 1: Imputation results on UCI datasets - RMSE (lower is better).

| Data | MCAR | | | MAR | | |
|---|---|---|---|---|---|---|
| | EMFlow | MCFlow | GAIN | EMFlow | MCFlow | GAIN |
| News | **.139 ± .001** | .167 ± .0015 | .197 ± .005 | **.172 ± .000** | .181 ± .001 | .271 ± .039 |
| Air | **.097 ± .005** | .111 ± .004 | .127 ± .005 | **.040 ± .001** | .055 ± .001 | .061 ± .023 |
| Letter | **.111 ± .001** | .121 ± .000 | .127 ± .001 | **.110 ± .001** | .127 ± .001 | .166 ± .040 |
| Concrete | **.147 ± .004** | .233 ± .007 | .194 ± .008 | **.133 ± .006** | .198 ± .010 | .184 ± .012 |
| Review | **.229 ± .003** | .234 ± .004 | .278 ± .005 | .194 ± .004 | **.191 ± .004** | .234 ± .014 |
| Credit | **.125 ± .001** | .135 ± .002 | .131 ± .002 | **.024 ± .001** | .028 ± .002 | .029 ± .002 |
| Energy | **.086 ± .001** | .092 ± .001 | .110 ± .002 | **.175 ± .002** | .176 ± .002 | .250 ± .019 |
| CTG | **.104 ± .006** | .140 ± .005 | .143 ± .008 | **.105 ± .001** | .153 ± .003 | .165 ± .006 |
| Song | **.025 ± .000** | .030 ± .006 | .034 ± .002 | **.024 ± .000** | .031 ± .014 | .028 ± .001 |
| Wine | **.076 ± .001** | .098 ± .002 | .097 ± .002 | **.102 ± .002** | .124 ± .003 | .135 ± .007 |
| Web | **.001 ± .000** | .002 ± .000 | .003 ± .003 | **.002 ± .004** | .006 ± .009 | .006 ± .010 |

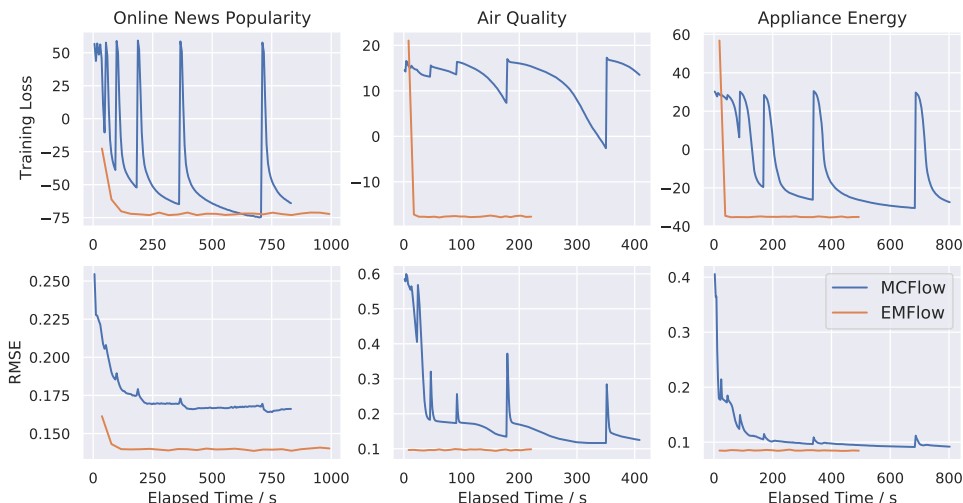

Figure 2: Comparison of convergence speed in terms of the training loss and test set RMSE on three UCI datasets.

## 4 EXPERIMENTS

In this section, we evaluate the performance of EMFlow on multivariate and image datasets in terms of the imputation quality and and the speed of model training. Its performance is compared to that of MCFlow, the most related competitor that has been shown to be superior to other state of art methods (Richardson et al., 2020).

Table 2: Imputation results on image datasets - RMSE (lower is better)

|  | Missing Rate | .1 | .2 | .3 | .4 | .5 | .6 | .7 | .8 | .9 |
|---|---|---|---|---|---|---|---|---|---|---|
| MNIST | GAIN | .1029 | .1184 | .1399 | .1495 | .1723 | .1794 | .2167 | .2200 | .2710 |
|  | MisGAN | .1083 | .1117 | .1184 | .1227 | .1311 | .1388 | .1512 | .1906 | .2621 |
|  | MCFlow | .0835 | .0879 | .0894 | .0941 | .1027 | .1119 | **.1251** | **.1463** | .2020 |
|  | EMFlow | **.0726** | **.0775** | **.0832** | **.0901** | **.0986** | **.1100** | .1260 | .1504 | **.1951** |
| CIFAR-10 | GAIN | .1025 | .1090 | .1103 | .1073 | .1094 | .1202 | .1217 | .1426 | .5388 |
|  | MisGAN | .1577 | .1434 | .1478 | .1326 | .1588 | .1824 | .2036 | .2660 | .3011 |
|  | MCFlow | .1083 | .1112 | .1179 | .1273 | .1340 | .1387 | .1466 | .1552 | .1702 |
|  | EMFlow | **.0444** | **.0479** | **.0525** | **.0575** | **.0619** | **.0689** | **.0782** | **.0926** | **.1188** |

Table 3: Classification accuracy on imputed image datasets (higher is better)

|  | Missing Rate | .1 | .2 | .3 | .4 | .5 | .6 | .7 | .8 | .9 |
|---|---|---|---|---|---|---|---|---|---|---|
| MNIST | MCFlow | **.9894** | .9878 | .9878 | .9871 | .9840 | .9806 | .9659 | **.9331** | **.7732** |
|  | EMFlow | **.9894** | **.9884** | **.9882** | **.9878** | **.9860** | **.9824** | **.9696** | .9253 | .7502 |
| CIFAR-10 | MCFlow | .8352 | .7081 | .5525 | .4166 | .3406 | .2820 | .2476 | .2194 | .1875 |
|  | EMFlow | **.9085** | **.8974** | **.8783** | **.8535** | **.8116** | **.7446** | **.6214** | **.4868** | **.3127** |

To make the comparison more objective, both models use the same normalizing flow with six affine coupling layers. We also follow the authors' suggestion for the the hyperparameter selection of MCFlow throughout this section. Additionally, we also present the benchmarks of other state-of-art models including GAIN (Yoon et al., 2018) and MisGAN (Li et al., 2019) for more comprehensive comparisons.

## 4.1 MULTIVARIATE DATASETS

Ten multivariate datasets from the UCI repository (Dheeru & Taniskidou, 2017) are used for evaluation. For all of them, each feature is scaled to fit inside the interval $[0, 1]$ via min-max normalization. We simulate MCAR with a missing rate of 0.2 by removing each value independently according to a Bernoulli distribution. We also simulate MAR scenario where the missing probability of the last 30% features depends on the values of the first 70% features[3].

The initial imputation is performed by randomly sampling from the observed entries of each feature. All experiments are conducted using five-fold cross validation where the test set only goes through the re-imputation phase in each iteration. The choices of hyperparameters are detailed in appendix F, where we also show that EMFlow is not sensitive to the choice of hyperparameters.

**Results** The imputation performace is evaluated by calculating the Root Mean Squared Error (RMSE) between the imputed and true values. As shown in Table 1, EMFlow performs constantly better than MCFlow under both MCAR and MAR settings for nearly all datasets.

Additionally, We trained EMFlow and MCFlow on the same machine with the same learning rate and batch size to compare the convergence speed. Figure 2 shows the training loss and the test set RMSE over time on three UCI datasets. It shows that EMFlow converges significantly faster than MCFlow. In fact, EMFlow converges within three iterations for most of the UCI datasets.

## 4.2 IMAGE DATASETS

We also evaluate EMFlow on MNIST and CIFAR-10. MNIST is a dataset of 28×28 grayscale images of handwritten digits (LeCun et al., 1998), and CIFAR-10 is a dataset of 32×32 colorful images from 10 classes (Krizhevsky et al., 2009). For both datasets, the pixel values of each image are scaled to

---

[3]See appendix E for the details of how MAR is simulated.

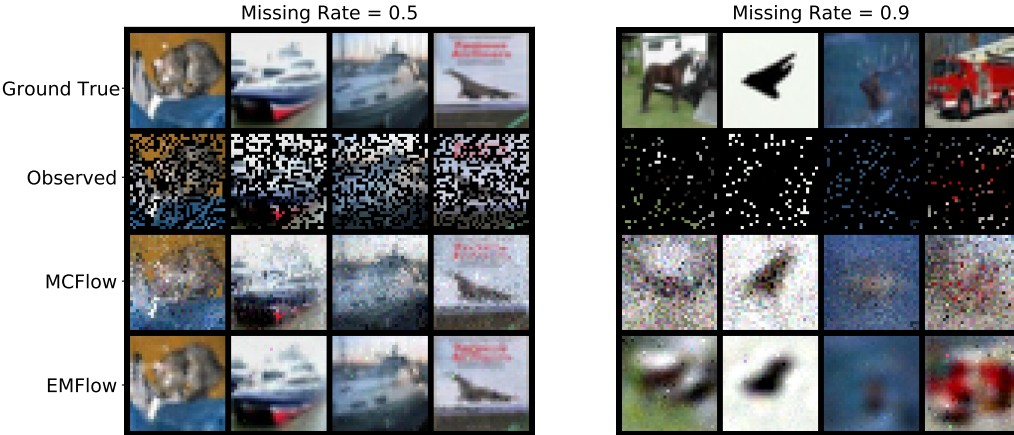

Figure 3: Sample imputed images for CIFAR-10 at missing rates of 0.5 and 0.9.

$[0, 1]$. In this section, we simulate MCAR where each pixel is independently missing with various probabilities from 0.1 to 0.9.

The initial imputation is performed by nearest-neighbor sampling where a missing pixel is filled by one of its nearest observed neighbors. In our experiments, the standard 60,000/10,000 and 50,000/10,000 training-test set partitions are used for MNIST and CIFAR-10 respectively. The choices of hyperparameters are detailed in appendix F.

**Results**    Table 2 shows the RMSE of all considered methods on both image datasets. In the case of MNIST, EMFlow and MCFlow have similar RMSE and outperform other methods, while MCFlow starts to gain slight advantage under high missing rates. In the case of CIFAR-10, EMFlow achieve much lower RMSE than all competing methods.

To further demonstrate the efficiency of EMFlow, We also compare EMFlow and MCFlow with respect to the accuracy of post-imputation classification. For this purpose, a LeNet-based model and a VGG19 model were trained on the original training sets of MNIST and CIFAR-10 respectively. These models then made predictions on the imputed test sets under different missing rates. Table 3 shows that EMFlow yields slightly better post-imputation prediction accuracy than MCFlow on MNIST, while the improvement is much more significant on CIFAR-10. We note that these findings are in good agreement with the RMSE results.

To qualitatively compare the imputation quality of EMFlow and MCFlow, Figure 3 shows sample imputed images from CIFAR-10 with MCAR missing rates at 0.5 and 0.9. The first row includes the (complete) ground truth images for reference, while the second row includes the (incomplete) observed images on which the models were trained. The last two rows showcase the reconstructed images by MCFlow and EMFlow, respectively. It's clear that EMFlow performs better than MCFlow by recovering more details and displaying sharper boundaries and cleaner background.

## 5    CONCLUSION

We propose a novel architecture EMFlow for missing data imputation. It combines the strength of the online EM and the normalizing flow to learn the density estimation in the presence of incomplete data while performing imputation. Various experiments with multivariate and image datasets show that EMFlow significantly outperforms its state-of-art competitor with respect to imputation accuracy as well as the convergence speed under a wide range of missing rates and different missing mechanisms. The accuracy of post-imputation classification on image datasets also demonstrates the superior EMFlow's ability of recovering semantic structure from incomplete data.

### REPRODUCIBILITY STATEMENT

The code and data to reproduce the results in this work are included in the supplementary materials.

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

## A  AFFINE COUPLING LAYERS

The basic idea behind normalizing flow is to find a transformation (of $X$) that would be able to represent a complex density function as a function of simpler ones, such as a multivariate Gaussian random variable (to be denoted by $Z$). The quest for an invertible transformation $f$, such that $Z = f^{-1}(X) \sim N_p(0, I)$ can be challenging, but in theory such a transformation does exist. Brockwell (2007) showed the existence of such a transformation which converts any $p$-dimensional random variable $X$ to $p$ independent uniform random variables $U_j = g_j(X) \sim U(0, 1)$ for $j = 1, \ldots, p$, which are known as universal residuals. Thus, choosing $Z_j = f_j^{-1}(X) \equiv \Phi^{-1}(g_j(X)) \sim N(0, 1)$, where $\Phi(\cdot)$ denotes the cumulative distribution function of a standard normal distribution, we can show the existence of $Z = f(X)$, where $f^{-1}(x) = (f_1^{-1}(x), \ldots, f_p^{-1}(x))$ that enables to transform $X$ to $Z$. However, finding such nonlinear transform based on observed data is challenging and so we use sequence of simpler transforms in the spirit of universal residuals, for simplicity and computational efficiency.

Following the framework of Real NVP (Dinh et al., 2016), the NF used throughout this work consists of a sequence of affine coupling layers $f_{\text{aff}} : \mathbf{x} \mapsto \mathbf{y}$ defined as

$$
\begin{aligned}
\mathbf{y}_{1:d} &= \mathbf{x}_{1:d} \\
\mathbf{y}_{d+1:p} &= \mathbf{x}_{d+1:p} \odot \exp\left(s\left(\mathbf{x}_{1:d}\right)\right) + t\left(\mathbf{x}_{1:d}\right)
\end{aligned}
\tag{16}
$$

where the scale and shift parameters $s(\cdot)$ and $t(\cdot)$ are usually implemented by neural networks, and $\odot$ is the element-wise product.

Therefore, the input $\mathbf{x}$ are split into two parts: the first $d$ dimensions stay the same, while the other dimensions undergo an affine transformation whose parameters are the functions of the first $d$ dimensions.

## B  ONLINE EM FOR MISSING DATA IMPUTATION

In this section, the steps of vanilla EM in the context of missing data imputation are reviewed. We then show how the online EM framework proposed by Cappé & Moulines (2009) can be easily applied here.

### B.1  VANILLA EM

Given $n$ i.i.d data points $\{\mathbf{z}_i = (\mathbf{z}_i^o, \mathbf{z}_i^m)\}_{i=1}^n$ distributed under $\mathcal{N}(\mu, \Sigma)$, the E-step of each EM iteration evaluates the conditional expectation of the complete data likelihood:

$$Q(\phi; \widehat{\phi}^{(t)}) = \frac{1}{n} \sum_{i=1}^n Q_i(\phi; \widehat{\phi}^{(t)}) = \frac{1}{n} \sum_{i=1}^n E_{\widehat{\phi}^{(t)}} \left[ \log \mathcal{N}(\mathbf{z}_i; \phi) | \mathbf{z}_i^o \right] \tag{17}$$

where $\phi = (\boldsymbol{\mu}, \boldsymbol{\Sigma})$, and $\widehat{\phi}^{(t)} = (\widehat{\boldsymbol{\mu}}^{(t)}, \widehat{\boldsymbol{\Sigma}}^{(t)})$ are the estimates at the $t^{th}$ iteration. We also follow the treatment by Cappé & Moulines (2009) to normalize the conditional expectation by $1/n$ for easier transition to the online version of EM.

To calculate conditional expectation inside the summation, we can use the fact that the conditional distribution of $\mathbf{z}_i^m$ given $\mathbf{z}_i^o$ is still Gaussian:

$$\mathbf{z}_i^m | \mathbf{z}_i^o; \widehat{\phi}^{(t)} \sim \mathcal{N}(\widetilde{\boldsymbol{\mu}}_i^{m(t)}, \widetilde{\boldsymbol{\Sigma}}_i^{m(t)}) \tag{18}$$

where

$$\begin{aligned}
\widetilde{\boldsymbol{\mu}}_i^{m(t)} &= \widehat{\boldsymbol{\mu}}_{\mathbf{m}_i}^{(t)} + \widehat{\boldsymbol{\Sigma}}_{\mathbf{m}_i \mathbf{o}_i}^{(t)} \left( \widehat{\boldsymbol{\Sigma}}_{\mathbf{o}_i \mathbf{o}_i}^{(t)} \right)^{-1} \left( \mathbf{z}_i^o - \widehat{\boldsymbol{\mu}}_{\mathbf{o}_i}^{(t)} \right), \\
\widetilde{\boldsymbol{\Sigma}}_i^{m(t)} &= \widehat{\boldsymbol{\Sigma}}_{\mathbf{m}_i \mathbf{m}_i}^{(t)} - \widehat{\boldsymbol{\Sigma}}_{\mathbf{m}_i \mathbf{o}_i}^{(t)} \left( \widehat{\boldsymbol{\Sigma}}_{\mathbf{o}_i \mathbf{o}_i}^{(t)} \right)^{-1} \widehat{\boldsymbol{\Sigma}}_{\mathbf{o}_i \mathbf{m}_i}^{(t)},
\end{aligned} \tag{19}$$

$\mathbf{m}_i$ and $\mathbf{o}_i$ are the missing and observed masks respectively, and the subscripts of $(\widehat{\boldsymbol{\mu}}^{(t)}, \widehat{\boldsymbol{\Sigma}}^{(t)})$ represent the slicing indexes.

Then it's easily easy to find

$$\begin{aligned}
Q_i(\phi; \widehat{\phi}^{(t)}) = &-\frac{1}{2} \left[ \left( \widehat{\mathbf{z}}_i^{(t)} - \boldsymbol{\mu} \right)^T \boldsymbol{\Sigma}^{-1} \left( \widehat{\mathbf{z}}_i^{(t)} - \boldsymbol{\mu} \right) \right. \\
&+ \mathrm{Tr} \left( \left[ \left( \boldsymbol{\Sigma}^{-1} \right)_{\mathbf{m}_i \mathbf{m}_i} \right] \widetilde{\boldsymbol{\Sigma}}_i^{m(t)} \right) \right] - \frac{1}{2} \log |2\pi\boldsymbol{\Sigma}|
\end{aligned} \tag{20}$$

where $\mathrm{Tr}(\cdot)$ denotes the matrix trace, and $\widehat{\mathbf{z}}_i^{(t)}$ is in fact the imputed data vector whose missing part is replaced by the conditional mean $\widetilde{\boldsymbol{\mu}}_i^{m(t)}$.

When it comes to the M-step, the derivatives of $Q(\phi; \phi^{(t)})$ are calculated as

$$\begin{aligned}
\frac{\partial Q}{\partial \boldsymbol{\mu}} &= \frac{1}{n} \boldsymbol{\Sigma}^{-1} \sum_{i=1}^n \left( \widehat{\mathbf{z}}_i^{(t)} - \boldsymbol{\mu} \right) \\
\frac{\partial Q}{\partial \boldsymbol{\Sigma}^{-1}} &= -\frac{1}{2n} \sum_{i=1}^n \left( \widehat{\mathbf{z}}_i^{(t)} - \boldsymbol{\mu} \right) \left( \widehat{\mathbf{z}}_i^{(t)} - \boldsymbol{\mu} \right)^T + \frac{1}{2} \left[ \boldsymbol{\Sigma} - \frac{1}{n} \sum_{i=1}^n \widetilde{\boldsymbol{\Sigma}}_i^{(t)} \right]
\end{aligned} \tag{21}$$

where $\widetilde{\boldsymbol{\Sigma}}_i^{(t)}$ is $p \times p$ matrix satisfying $\widetilde{\boldsymbol{\Sigma}}_{i,\mathbf{m}_i \mathbf{m}_i}^{(t)} = \widetilde{\boldsymbol{\Sigma}}_i^{m(t)}$ and all other elements equal to 0.

Therefore, the maximizer of $Q(\phi; \widehat{\phi}^{(t)})$ are

$$
\begin{aligned}
\widehat{\boldsymbol{\mu}}^{(t+1)} &= \frac{1}{n} \sum_{i=1}^{n} \widehat{\mathbf{z}}_i^{(t)}, \\
\widehat{\boldsymbol{\Sigma}}^{(t+1)} &= \frac{1}{n} \sum_{i=1}^{n} \left[ \left( \widehat{\mathbf{z}}_i^{(t)} - \boldsymbol{\mu}^{(t+1)} \right) \left( \widehat{\mathbf{z}}_i^{(t)} - \boldsymbol{\mu}^{(t+1)} \right)^T + \widetilde{\boldsymbol{\Sigma}}_i^{(t)} \right].
\end{aligned}
\tag{22}
$$

Here we provide a remark that helps to motivate the optimization of EMFlow described in Section 3.4.

**Remark.** *Given an incomplete data point* $\mathbf{z} = (\mathbf{z}^o, \mathbf{z}^m) \sim \mathcal{N}(\boldsymbol{\mu}, \boldsymbol{\Sigma})$, *its likelihood is maximized at* $\mathbf{z}^m = E\left[\mathbf{z}^m | \mathbf{z}^o\right]$.

To prove it, first note that the log-likelihood of $\mathbf{z}$ can be written as

$$
\log \mathcal{N}(\mathbf{z}^m, \mathbf{z}^o | \boldsymbol{\mu}, \boldsymbol{\Sigma}) = -\frac{1}{2} \left[ \begin{array}{c} \mathbf{z}^m - \boldsymbol{\mu}_{\mathbf{m}} \\ \mathbf{z}^o - \boldsymbol{\mu}_{\mathbf{o}} \end{array} \right]^{\mathrm{T}} \boldsymbol{\Sigma}^{-1} \left[ \begin{array}{c} \mathbf{z}^m - \boldsymbol{\mu}_{\mathbf{m}} \\ \mathbf{z}^o - \boldsymbol{\mu}_{\mathbf{o}} \end{array} \right] + C
\tag{23}
$$

Denote the conditional mean and covariance of $\mathbf{z}^m$ as $\widetilde{\boldsymbol{\mu}} = \boldsymbol{\mu}_{\mathbf{m}} + \boldsymbol{\Sigma}_{\mathbf{mo}} \left( \boldsymbol{\Sigma}_{\mathbf{oo}} \right)^{-1} (\mathbf{z}^o - \boldsymbol{\mu}_{\mathbf{o}})$ and $\widetilde{\boldsymbol{\Sigma}} = \boldsymbol{\Sigma}_{\mathbf{mm}} - \boldsymbol{\Sigma}_{\mathbf{mo}} \left( \boldsymbol{\Sigma}_{\mathbf{oo}} \right)^{-1} \boldsymbol{\Sigma}_{\mathbf{om}}$. The matrix block-wise inversion gives

$$
\boldsymbol{\Sigma}^{-1} = \left[ \begin{array}{cc} \boldsymbol{\Sigma}_{\mathbf{mm}} & \boldsymbol{\Sigma}_{\mathbf{mo}} \\ \boldsymbol{\Sigma}_{\mathbf{om}} & \boldsymbol{\Sigma}_{\mathbf{oo}} \end{array} \right]^{-1} = \left[ \begin{array}{cc} \boldsymbol{\Sigma}_{11}^{*} & \boldsymbol{\Sigma}_{12}^{*} \\ \boldsymbol{\Sigma}_{21}^{*} & \boldsymbol{\Sigma}_{22}^{*} \end{array} \right]
\tag{24}
$$

where

$$
\begin{aligned}
\boldsymbol{\Sigma}_{11}^{*} &= \widetilde{\boldsymbol{\Sigma}}^{-1} \quad \boldsymbol{\Sigma}_{12}^{*} = -\widetilde{\boldsymbol{\Sigma}}^{-1} \boldsymbol{\Sigma}_{mo} \boldsymbol{\Sigma}_{oo}^{-1} \\
\boldsymbol{\Sigma}_{21}^{*} &= -\boldsymbol{\Sigma}_{oo}^{-1} \boldsymbol{\Sigma}_{mo} \widetilde{\boldsymbol{\Sigma}}^{-1} \\
\boldsymbol{\Sigma}_{22}^{*} &= \boldsymbol{\Sigma}_{oo}^{-1} + \boldsymbol{\Sigma}_{oo}^{-1} \boldsymbol{\Sigma}_{mo} \widetilde{\boldsymbol{\Sigma}}^{-1} \boldsymbol{\Sigma}_{mo} \boldsymbol{\Sigma}_{oo}^{-1}.
\end{aligned}
\tag{25}
$$

After a series of linear algebra, we have

$$
\begin{aligned}
&\log \mathcal{N}(\mathbf{z}^m, \mathbf{z}^o | \boldsymbol{\mu}, \boldsymbol{\Sigma}) \\
&= -\frac{1}{2} \left( \mathbf{z}^m - \widetilde{\boldsymbol{\mu}} \right)^{\mathrm{T}} \widetilde{\boldsymbol{\Sigma}}^{-1} \left( \mathbf{z}^m - \widetilde{\boldsymbol{\mu}} \right) - \frac{1}{2} \left( \mathbf{z}^o - \boldsymbol{\mu}_o \right)^{\mathrm{T}} \boldsymbol{\Sigma}_{oo}^{-1} \left( \mathbf{z}^o - \boldsymbol{\mu}_o \right) + C
\end{aligned}
\tag{26}
$$

As $\widetilde{\boldsymbol{\Sigma}}^{-1}$ is positive definite, it is obvious that the likelihood reaches the maximum at $\mathbf{z}^m = \widetilde{\boldsymbol{\mu}}$.

## B.2 ONLINE EM

It's clear that the EM algorithm needs to process the whole dataset at each iteration, which becomes impractical when dealing with huge datasets. To address this problem, Cappé & Moulines (2009) propose to replace the reestimation functional in the E-step by a stochastic approximation step:

$$
\hat{Q}^{(t+1)}(\phi) = (1 - \rho_{t+1}) \hat{Q}^{(t)}(\phi) + \rho_{t+1} \cdot \frac{1}{|B|} \sum_{i \in B} E_{\widehat{\phi}^{(t)}} \left[ \log f\left( \mathbf{z}_i; \phi \right) \mid \mathbf{z}_i^o \right]
\tag{27}
$$

where $f$ is the probability density of complete data, $\{\rho_i\}_{i=1}^{\infty}$ are a series of step sizes satisfying the conditions specified in equation 9, $B \subset \{1, \ldots, n\}$ denotes the indexes of a mini-batch of samples, and $|B|$ is the batch size.

Under some mild regulations such as the complete data model belongs to exponential family Cappé & Moulines (2009), equation 27 boils down to a stochastic approximation of the estimates of sufficient statistics:

$$\widehat{s}^{(t+1)} = (1 - \rho_{t+1})\widehat{s}^{(t)} + \rho_{t+1}\frac{1}{|B|}\sum_{i \in B} E_{\widehat{\phi}^{(t)}}\left[S(\mathbf{z})|\mathbf{z}_i^o\right] \tag{28}$$

where $S(\mathbf{z})$ is the sufficient statistic of $f$.

Note that the sufficient statistics for multivariate Gaussian are just sample mean and sample covariance. Therefore, equation 8 follows immediately form equation 28.

## C    Interpret the Optimization

As described in Section 3.1, the estimated model parameters and the imputed data are supposed to maximize the complete data likelihood (only one data point is considered here for illustration purpose), namely,

$$(\widehat{\mathbf{x}}, \widehat{\psi}, \widehat{\boldsymbol{\mu}}, \widehat{\boldsymbol{\Sigma}}) = \underset{\mathbf{x} \in \mathcal{X}', \psi, \boldsymbol{\mu}, \boldsymbol{\Sigma}}{\arg\max} \log p_X(\mathbf{x}|\psi, \boldsymbol{\mu}, \boldsymbol{\Sigma}) \tag{29}$$

While it's hard to optimize such objective simultaneously, it's possible to optimize it alternatively. For example, learning $L_1(\psi)$ in equation 11 corresponds to updating $\psi$ while keeping other variables unchanged.

The operations performed in the latent space needs more explanation. First of all, it is shown in appendix B that the imputation increases the complete likelihood in the latent space, i.e.,

$$\mathcal{N}(\widehat{\mathbf{z}}; \widehat{\boldsymbol{\mu}}, \widehat{\boldsymbol{\Sigma}}) \geq \mathcal{N}(\mathbf{z}; \widehat{\boldsymbol{\mu}}, \widehat{\boldsymbol{\Sigma}}) \tag{30}$$

where $\mathbf{z}$ and $\widehat{\mathbf{z}}$ are the embedding vectors before and after the imputation, respectively.

However, the increase of likelihood in the latent space doesn't guarantee the same thing in the data space. Therefore, the flow model is updated again by optimizing $L_2(\psi)$ in equation 14 to ensure that $\widetilde{\mathbf{x}} = f_\psi(\widehat{\mathbf{z}})$ has a high likelihood in the data space $\mathcal{X}$. Note that although $\widetilde{\mathbf{x}}$ doesn't necessarily belong to $\mathcal{X}'$, the reconstruction penalty $L_{rec}$ defined in equation 15 forces $\widetilde{\mathbf{x}}$ to be close to $\mathcal{X}'$. Finally, in the re-imputation phase, the imputed data $\widehat{\mathbf{x}}$ are updated by projecting $\widetilde{\mathbf{x}}$ onto $\mathcal{X}'$.

## D    Implementation Details

**Model Initialization**    Each inference iteration consists of a training phase and a re-imputation phase. And since the latter phase updates the current imputation, the parameters of the flow model $f_\psi$ are reinitialized after each iteration to learn the new data density faster.

The base distribution estimate $\mathcal{N}(\mathbf{z}; \widehat{\boldsymbol{\mu}}, \widehat{\boldsymbol{\Sigma}})$ is initialized at the very beginning when the online EM encounters the first batch of embedding vectors. Specifically $\widehat{\boldsymbol{\mu}}$ and $\widehat{\boldsymbol{\Sigma}}$ are initialized to be the sample mean and sample covariance. By default, the base distribution is also reinitialized along with the flow model after each iteration.

**Stabilization of Online EM**    Online EM can be unstable in some cases where the covariance estimate $\widehat{\Sigma}$ becomes ill-conditioned during the inference. Such instability can be traced back to two primary sources: (i) the initial naive imputation has really bad quality and the inter-feature dependency is distorted significantly; (ii) the batch size is close to or less than the number of features, which makes it hard to estimate the covariance directly (e.g. Fan et al., 2008; Chen et al., 2011).

Fortunately, those two issues can be solved easily. To reduce the impact of the initial naive imputation and make the covariance estimation more robust, we enlarge the diagonal entries of the original covariance estimate from equation 8 proportionally:

$$\widehat{\boldsymbol{\Sigma}}^{Rob} = \widehat{\boldsymbol{\Sigma}} + \beta \cdot \text{Diag}(\widehat{\boldsymbol{\Sigma}}) \tag{31}$$

where $\mathrm{Diag}(\widehat{\boldsymbol{\Sigma}})$ is a diagonal matrix with the same diagonal entries as $\widehat{\boldsymbol{\Sigma}}$, and $\beta$ a positive hyper-parameter. In practice, $\beta$ would be decreased gradually to 0 during the first a few iterations as the quality of current imputation gets better and better.

To address the second issue, the training phase in Algorithm 1 can be modified to make the online EM *see* more data points. When updating the base distribution, the current batch of embedding vectors can be concatenated with previous ones to form a *super*-batch with a maximum size $S_{\mathrm{super}}$. If the maximum size is exceeded, the most previous embedding vectors would be excluded. And since the previous batches are already imputed, the super-batch brings nearly no additional computational overhead.

## E    SIMULATE MAR ON UCI DATASETS

To simulate MAR, the first 70% features of each data point $\mathbf{x}_i$ is retained, and the remaining 30% features are removed with probability:

$$\mathrm{sigmoid}\left( \sum_{j=1}^{\lfloor 0.7p \rfloor} x_{ij} \right) \tag{32}$$

where $\lfloor \cdot \rfloor$ denotes the integer part of a number.

Note that different data sets would exhibit different missing rates under our MAR setting. Table shows the dimensions of UCI data sets used in our experiment as well as their MAR missing rates for the last 30% features.

Table 4: Information of UCI datasets.

| Data | Features | Samples | MAR missing rate |
|---|---|---|---|
| News | 60 | 39644 | 0.35 |
| Air | 13 | 9357 | 0.62 |
| Letter | 16 | 19999 | 0.57 |
| Concrete | 9 | 1030 | 0.50 |
| Review | 24 | 5454 | 0.49 |
| Credit | 23 | 30000 | 0.30 |
| Energy | 28 | 19735 | 0.39 |
| CTG | 21 | 2126 | 0.29 |
| Song | 90 | 92743 | 0.32 |
| Wine | 12 | 6497 | 0.26 |
| Web | 550 | 58638 | 0.12 |

## F    HYPERPARAMETER ROBUSTNESS

EMFlow has a relatively simpler architecture than GAN-based models, and there are just two most important two hyperparameters specific to it. One is the power index $\gamma$ in the step size schedule of online EM (see equation 10), while the other is the strength of the reconstruction error $\alpha$ in the objective (see equation 14). From our extensive experiments, $\gamma \in [0.6, 0.9]$ often leads to consistent results. On the other hand, a practical guide for choosing $\alpha$ is to make the magnitudes of first and second term of equation 14 don't differ too much.

Figure 4 shows RMSE on the test set of Online News Popularity dataset versus different values of $\gamma$ and $\alpha$, while all other hyperparameters are kept fixed. The RMSEs in all configurations are well below that obtained by MCFlow, and their fluctuations are almost negligible.

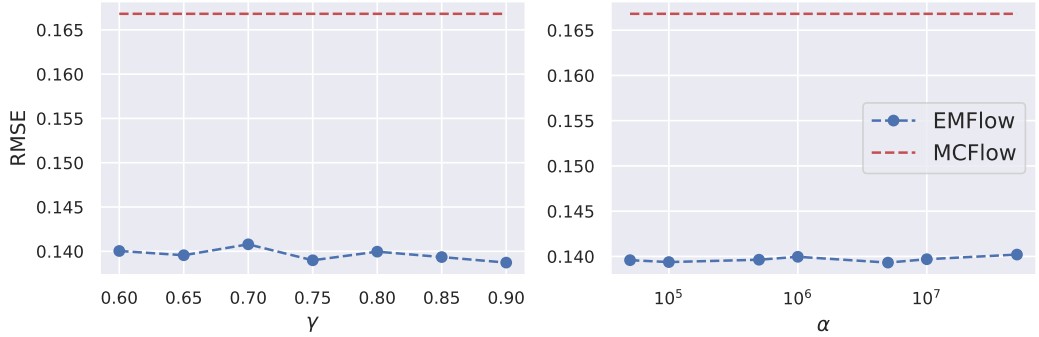

Figure 4: RMSE between true and imputed values on the test set of Online News Popularity dataset with different choices of $\gamma$ and $\alpha$.

### F.1 HYPERPARAMETERS FOR UCI DATASETS

We use $\alpha = 10^6$ and choose $\rho_t = 0.99 \cdot t^{-0.8}$ as the step size schedule for all UCI datasets. During the training of EMFlow, the batch size is 256 and the learning rate is $1 \times 10^{-4}$. Compared to MCAR, the initial imputation can be more difficult under MAR where the marginal observed density is distorted more. Therefore, the robust covariance estimation in equation 31 is adopted by default under MAR. Specifically, we use $\beta = 10^{-2}$ for the first two iterations, $\beta = 10^{-3}$ for the next two iterations, and $\beta = 0$ for the remaining ones.

### F.2 HYPERPARAMETERS FOR IMAGE DATASETS

For all experiments with the image datasets, the step size schedule of online EM is again $\rho_t = 0.99 \cdot t^{-0.8}$. For MNIST and CIFAR-10, we choose $\alpha$ to be $5 \times 10^8$ and $1 \times 10^6$ respectively. During the training of EMFlow, the batch size is 512 and the learning rate is $1 \times 10^{-3}$. Since image datasets have much larger dimensions than UCI datasets, the super-batch approach with a size of 3000 is adopted. Additionally, similar robust covariance estimation used by UCI datasets is also applied to CIFAR-10.

## G   ADDITIONAL EXPERIMENT

### G.1   MODEL CONGENIALITY

We also consider the congeniality of the imputation model that measures its ability to preserve the feature-label relationship (Meng, 1994; Burgess et al., 2013). To quantify how the feature-label relationship is changed after imputation, we calculated the euclidean distance between the coefficients of two linear or logistic regression models, one learned from a complete test set and the other learned from the corresponding imputed version. Three datasets with continuous or discrete targets were used in our experiments, and the results are shown in Table 5. It shows that the gaps in the measured congeniality actually coincide with those in the measured RMSE.

Table 5: Congeniality of imputation models (lower is better).

|  | Credit | Letter | News |
|---|---|---|---|
| GAIN | $1.3310 \pm 0.1378$ | $4.2957 \pm 0.2212$ | $0.0575 \pm 0.0762$ |
| MCFlow | $1.1080 \pm 0.1363$ | $3.8157 \pm 0.2580$ | $0.0541 \pm 0.0699$ |
| EMFlow | $\mathbf{0.9141 \pm 0.1112}$ | $\mathbf{3.7611 \pm 0.1910}$ | $\mathbf{0.0229 \pm 0.0240}$ |

Table 6: Imputation results of MIWAE on UCI datasets - RMSE (lower is better).

| Data | MCAR | | MAR | |
|---|---|---|---|---|
| | EMFlow | MIWAE | EMFlow | MIWAE |
| News | **.139 ± .001** | .203 ± .004 | **.172 ± .000** | .195 ± .005 |
| Air | **.097 ± .005** | .101 ± .008 | .040 ± .001 | **.034 ± .001** |
| Letter | .111 ± .001 | **.078 ± .001** | .110 ± .001 | **.085 ± .001** |
| Concrete | **.147 ± .004** | .151 ± .009 | **.133 ± .006** | .138 ± .009 |
| Review | **.229 ± .003** | .234 ± .007 | .194 ± .004 | **.192 ± .007** |
| Credit | **.125 ± .001** | .130 ± .001 | **.024 ± .001** | .026 ± .002 |
| Energy | **.086 ± .001** | .101 ± .011 | .175 ± .002 | **.167 ± .002** |
| CTG | .104 ± .006 | **.102 ± .005** | **.105 ± .001** | .108 ± .002 |
| Song | **.025 ± .000** | .031 ± .001 | **.024 ± .000** | .025 ± .000 |
| Wine | .076 ± .001 | **.072 ± .002** | .102 ± .002 | **.089 ± .002** |
| Web | **.0006 ± .0003** | .0022 ± .0010 | **.0022 ± .0042** | .0066 ± .0109 |

## G.2 COMPARISON TO MIWAE

In this section, we compare EMFlow with MIWAE that introduces the importance-weighted autoencoder for imputation tasks. For UCI datasets, we follow the implementation provided by the author [4]. On the other hand, the implementation of MIWAE on image datasets are not provided, and thus we tried our best to tune MIWAE on MNIST.

As shown in Table 6, EMFlow and MIWAE have mixed performance on UCI datasets, and the differences in most cases are quite small. Table 7 and 8 shows that EMFlow outperforms MIWAE on MNIST in terms of the RMSE on test set and post-imputation accuracy. We also note that EMFlow still converges much faster than MIWAE.

Table 7: Imputation results of MIWAE on MNIST - RMSE (lower is better)

| | Missing Rate | .1 | .2 | .3 | .4 | .5 | .6 | .7 | .8 | .9 |
|---|---|---|---|---|---|---|---|---|---|---|
| MNIST | MIWAE | .1022 | .1057 | .1120 | .1185 | .1194 | .1233 | .1409 | .1672 | .2275 |
| | EMFlow | **.0726** | **.0775** | **.0832** | **.0901** | **.0986** | **.1100** | **.1260** | **.1504** | **.1951** |

Table 8: Classification accuracy of MIWAE on imputed MNIST (higher is better)

| | Missing Rate | .1 | .2 | .3 | .4 | .5 | .6 | .7 | .8 | .9 |
|---|---|---|---|---|---|---|---|---|---|---|
| MNIST | MIWAE | .9792 | .9878 | .9683 | .9672 | .9521 | .9469 | .9237 | .9331 | .7005 |
| | EMFlow | **.9894** | **.9884** | **.9882** | **.9878** | **.9860** | **.9824** | **.9696** | **.9253** | **.7502** |

## G.3 EFFECT OF INITIAL IMPUTATION

EMFlow is an iterative imputation framework, and the nearest-neighbor (NN) imputation is performed for image datasets at the beginning as warm-start in all the previous experiments. In this section, we compare NN imputation and zero imputation as the starting point of EMFlow and shows the convergence of the training loss, the RMSE on the test set, and the Frobenius norm of the covariance matrix in the latent space. As shown in Figure, 5, the difference between NN imputation and zero imputation is negligible and the convergence under both cases is fast.

---

[4] https://github.com/pamattei/miwae

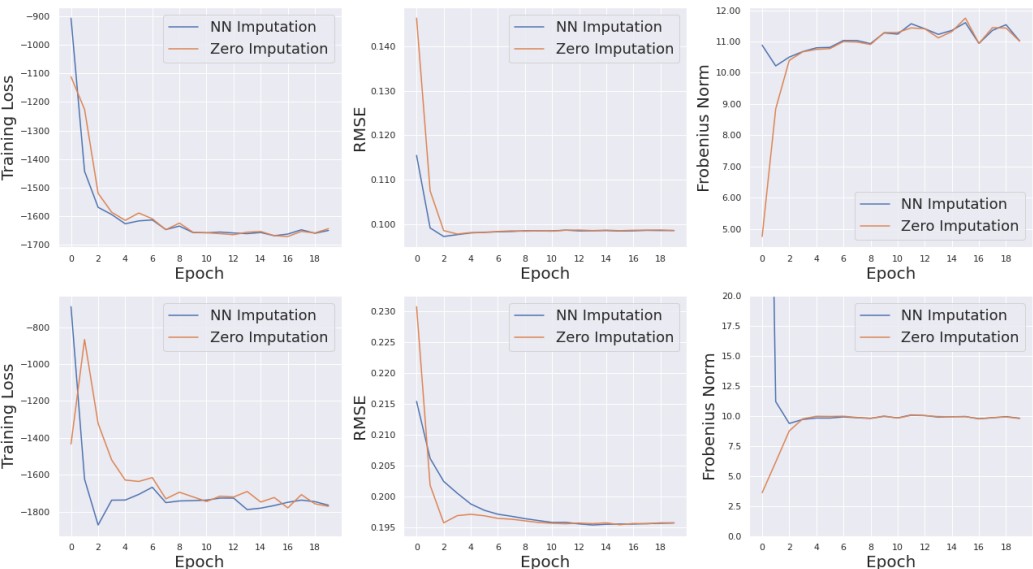

Figure 5: The convergence of training loss, test set RMSE and Frobenius norm of the latent covariance matrix using NN or zero imputation as the the start on MNIST. The first and second rows of plots correspond to the missing rates of 0.5 and 0.9.

