# OpenReview forum: "EMFlow: Data Imputation in Latent Space via EM and Deep Flow Models"
_ICLR.cc/2022/Conference — ICLR 2022 Submitted_

### Official Review · Reviewer_zdAh · 2021-10-20

**Correctness:** 4
**Technical Novelty And Significance:** 3
**Empirical Novelty And Significance:** 3
**Recommendation:** 6
**Confidence:** 5

**Main Review:**

The study is well designed. however, there are few limitations that needs to be acknowledged.

The data structure is using imaging data as example. This kind of dataset having high neighborhood correlation and imputation using EM with latent space will be less challenging comparing to time series dataset which also have additional layers of correlation between time points. Perhaps need to show the performance on those datasets.

The author did not show the performance in MNAR. Therefore the novelty remains limited.

The author also need to realize the limitation for all EM method comparing to FCS multiple imputation which considering the uncertainty of the imputed value.

**Summary Of The Paper:**

The authors propose a novel architecture EMFlow for missing data imputation. The authors also show the results of various experiments with multivariate and image datasets. Finally, the authors report the accuracy of post-imputation classification on image datasets.

**Summary Of The Review:**

The study and the expected results are acceptable; however, this framework was tested on imaging dataset, which has unique characteristics. for example, one can not assume similar performance when the method is tested on longitudinal clinical data, unless tested systematically on actual data.

---

> ### Author Response · Authors · 2021-11-21
> **Response to reviewer zdAh**
>
> We’re glad you had a positive impression on our paper. Please find our point-wise responses below to your insightful feedback.
>
> 1. __"time series dataset which also have additional layers of correlation between time points"__
>
>     The [Air(Quality) dataset](https://archive.ics.uci.edu/ml/datasets/Air+quality) included in our experiments is itself a time series dataset. But we admit that we didn't test EMFlow systematically on time series datasets. We understand that EMFlow and all other competing methods have an implicit assumption that each observation is independently drawn from an underlying distribution, and thus don't account for the correlations between between time points. To mitigate this issue, we think there might be at least two solutions to explore:
>     * Stack some lagged observations and apply EMFlow (like what is done for fitting AR models).
>     * Develop NFs that estimate time-dependent distributions, i.e., include time as an additional dimension in the data and latent space. There is already some interesting works in this area [1].
>
>     These are definitely worth pursuing as possible extensions of our proposed EMflow algorithm that accounts for correlation across the vector of observations.
>
> 2. __"The author did not show the performance in MNAR. Therefore the novelty remains limited"__
>
>     We admit that EMFlow is not designed for MNAR as we don't explicitly model the missing mechanism. We expect significant change needed for EMFlow to work on MNAR, like including another component to incorporate the domain knowledge of the missing process and learn the missing mechanism. It is well known that handling MNAR data requires strong modeling assumptions that depends on specific nature of data collection process and hence generic version like EMflow (applicable for MAR) or any other imputation methods would not work without domain knowledge. For this reason, we initially focused our application to MAR case for broader applications.
>
> 3. __"The author also need to realize the limitation for all EM method comparing to FCS multiple imputation which considering the uncertainty of the imputed value"__
>
>     It's true that EMFlow currently only does single imputation and thus the uncertainty of imputed values is missing. But it's possible to obtain the standard errors of the parameters (i.e., the covariance matrix) estimated by EM ([2], [3]), which could be potentially used for multiple imputation in the future work.
>
>     We also want to note that FCS methods normally need to specify a fully conditional distribution for each feature and sample from them sequentially in each iteration, which prevents FCS's application for high-dimensional data. However, it is also well known that FCs may not always uniquely determine a joint distribution or even lead to improper distributions if some regularity conditions (eg, as required for Hammersley-Clifford-Besag type results) are not met or not checked.
>
> 4. __"one can not assume similar performance when the method is tested on longitudinal clinical data, unless tested systematically on actual data"__
>
>     We added a longitudinal dataset ([Web](https://www.kaggle.com/c/web-traffic-time-series-forecasting)) for in our experiments in Table 1. Although it is not a clinical one and we don't have systematical test on such data, we believe that EMFlow can be applied to longitudinal clinical datasets as long as each subject is assumed to be independent.
>
> References
>
> [1] Both, Gert-Jan, and Remy Kusters. "Temporal Normalizing Flows." arXiv preprint arXiv:1912.09092 (2019).
>
> [2] Meng, Xiao-Li, and Donald B. Rubin. "Using EM to obtain asymptotic variance-covariance matrices: The SEM algorithm." Journal of the American Statistical Association 86.416 (1991): 899-909
>
> [3] Jamshidian, Mortaza, and Robert I. Jennrich. "Standard errors for EM estimation." Journal of the Royal Statistical Society: Series B (Statistical Methodology) 62.2 (2000): 257-270.

---

### Official Review · Reviewer_Twou · 2021-11-01

**Correctness:** 2
**Technical Novelty And Significance:** 2
**Empirical Novelty And Significance:** 2
**Recommendation:** 3
**Confidence:** 4

**Main Review:**

Although the imputation of MCAR and MAR scenarios has been studied for decades and in theory is not a problem, its applications in high-dimensional and real-world data with complicated distributions can be interesting from a practical perspective.

The work adapts the online EM algorithm for the missing data imputation. I quickly go through the online EM extension, which seems sound and no problem.

However, I have some concerns regarding the assumptions of NF.
It would be necessary to justify the assumptions, especially, why the two assumptions are reasonable to believe.

Firstly, inter-feature dependencies in the observation space are different from the inter-feature dependencies in the latent space. According to the studies in factor analysis and nonlinear ICA, the latent representation is not identifiable. So it is not straightforward to believe that the dependencies in the latent space are consistent with the dependencies in the observation space. Then, why would it be reasonable to do imputation in the latent space?

Secondly, note that according to the property of the change of variable formulation used by NF, the neighbors in the latent/source variable space are not necessary to be the neighbors in the observation space.
Another fact about NF is that even though the covariance matrix of NF is an identity matrix, it can still model the data distribution well. Then what are the dependencies modeled by the covariance of the multivariate Gaussian distribution in the latent space? Why we could believe that they are corresponding to inter-feature dependencies in the observed data?

A final minor concern is about the experiments. To have a better view of the comparison and the benefits of the proposed method, more comparison in the experiment section would be required, especially, the works using VAE and the methods without using generative models.


**Summary Of The Paper:**

The paper aims at imputing missing data which are MCAR and MAR.
For modeling the observed data distribution, it utilizes the framework of normalizing flow of which the latent variable/source variable space is Gaussian.
By assuming the consistency of inter-feature dependencies in the latent/source variable space, it applies online EM for the imputation of the latent space variables.
In the experiments, the proposed method, EMFlow is compared with GAIN, MisGAN, and MCFlow.

**Summary Of The Review:**

The work focuses on MCAR and MAR cases and extends NF and online EM for the missing data problem, which can be interesting for applications from a practical perspective. But the assumptions and main idea of the work can be lack justification, i.e., the relationship between the dependencies in the latent space and observation space needs to be elaborated and clarified. Moreover, a more thorough comparison with other related works would be helpful to better evaluate the proposed method.

---

> ### Author Response · Authors · 2021-11-21
> **Response to reviewer Twou**
>
> We thank the reviewer for taking time in reviewing our work, and  appreciate the in-depth comments on the assumptions of this work. However, we believe that there seem to be significant misunderstandings of our work in the review, as outlined below.
>
> 1. __"According to the studies in factor analysis and nonlinear ICA, the latent representation is not identifiable. So it is not straightforward to believe that the dependencies in the latent space are consistent with the dependencies in the observation space"__
>
>     It is to be noted that the NF framework is distinctly different from that of standard factor analysis, PCA or ICA and their nonlinear generalizations as the NF is built using a sequence of compositions of invertible maps. Thus, the identifiability issues are not confronted by NFs. Moreover, although its not straightforward to explicitly characterize dependence structure in the data space from those in the latent space (even though NF involves complicated invertible maps), the existence of such a map is well known (e.g, via the Rosenblatt transform).
>
> 2. __"the neighbors in the latent/source variable space are not necessary to be the neighbors in the observation space."__
>
>     The NF used in this work (Real NVP) performs element-wise transformation. That is, there is a univariable transformation between $x_i$ and $z_i$ for $i=1,\ldots,p$. So, the neighbors in the latent space are the neighbors in the data space.
>
> 3. __"Another fact about NF is that even though the covariance matrix of NF is an identity matrix, ..., Why we could believe that they are corresponding to inter-feature dependencies in the observed data?"__
>
>     There is a trade-off between the complexity of the base distribution and the expressiveness of the transformations in NF. For example, to model heavy-tailed distributions, either the base distribution should also be heavy-tailed or the transformation should be more expressive than Lipschitz-continuous affine transformations ([1]). That is, we can have the base distribution to capture some characteristics of the target distribution, rather than letting the transformations to do all the jobs. In this work, we deliberately assume a full covariance matrix in the latent space and which in turn allows it to capture the inter-feature dependency in the latent space. Considering that the transformations are element-wise, we further assume that the inter-feature dependencies in the latent and data spaces can be correlated. Such assumption is verified in the experiments in the sense that the imputed data vectors (transformed from imputed latent vectors) have reasonable accuracy.
>
> 4. __"more comparison in the experiment section would be required, especially, the works using VAE and the methods without using generative models"__
>
>     We added VAE-based MIWAE ([2]) as to our experiments and the results is shown in Appendix G.2. Since the authors of MIWAE didn't release the code on image datasets, we can only tune and test it on MNIST within this limited time frame. The results shows that EMFlow and MIWAE have very close results on UCI datasets, but EMFlow outperforms MIWAE on MNIST. We also note that EMFlow still converges much faster than MIWAE.
>
> References
>
> [1] Jaini, Priyank, et al. "Tails of lipschitz triangular flows." International Conference on Machine Learning. PMLR, 2020.
>
> [2] Mattei, Pierre-Alexandre, and Jes Frellsen. "MIWAE: Deep generative modelling and imputation of incomplete data sets." International Conference on Machine Learning. PMLR, 2019.

---

> > ### Comment · Reviewer_Twou · 2021-11-25
> > **A further elaboration and discussion**
> >
> > I would thank the authors for the response and would like to further discuss my main concerns. Let's see can we converge to the same page.
> >
> > I would like to further elaborate my point. The formulation of nonlinear ICA is: $x =f(s)$, where $x$ represents for the observed variables and $s$ represents for the noise/source signal. This is what NF is doing by using the change of variable formula: $p_x(x) = p_s(s)| det J_T (s)|^−1$.
> >
> > Given an observed vector $[x_1, x_2,...,x_m]$, one can get $[s_1, s_2 ,..., s_m]$ with the proposed method. My question is that without certain **identifiability**, how are the dimension information preserved, i.e., should $s_i$ be corresponding to and used for interpreting $x_i$? Furthermore, how the inter-feature dependencies are preserved, i.e., the dependencies between $X_i$ and $X_j$ imply the dependencies between $S_i$ and $S_j$? For example, one non-identifiable case can be that given $p(s)$, there is an equivalent class of $f$; another case can be that given $f$, $p(s)$ is invariant to any rotation of $s$.
> >
> > Moreover, I didn't get how Rosenblatt transform can deal with my concern regarding the dependencies and identifiability. I would sincerely ask the authors to further explain it for further discussion.

---

> > > ### Author Response · Authors · 2021-11-28
> > > **Further explanation on identifiability**
> > >
> > > We very much appreciate for your efforts to provide additional feedback with more elaboration. We hope you find our further explanation on identifiability satisfying.
> > >
> > > First of all, although nonlinear ICA and NF share some structural similarities, they have quite different goals: nonlinear ICA is primarily used for source identification and separation, while NF is primarily used to approximate the probability distribution of the data. We also note that recently [1] bridges the gap between NF and nonlinear ICA by developing a flow-based model for estimating latent representations with
> > > theoretical results on identifiability using equivalence classes.
> > >
> > > Due to the crucial distinction between the perceived goals of ICA and NF, identifiability of the parameters of the map is not necessarily the focus of this work. The primary goal of EMFlow is to perform imputation using posterior predictive distribution of missing values
> > > conditioned on the observed values using the NF to model the data distribution, and then use estimated predictive distribution on test data. Thus, identifiability of model parameters and their unique estimation may not be necessary. This is true for almost all NNs and so identifiability of parameters are usually ignored when predictions are the main goals. In fact, EMFlow only assumes a learnable correlation between the inter-feature-dependencies of the data and latent spaces, instead of coordinate correspondence (e.g., the dependencies between $X_i$ and $X_j$ imply the dependencies between $S_i$ and $S_j$ ).
> > >
> > > Empirically, such coordinate correspondence does exist for the current implementation of this work. A nice work to show such correspondence is [2] that concludes "there exists a correspondence between the coordinates in an image and in its learned
> > > representation" (see Figure 2 in section 5). However, as explained before, coordinate correspondence is not necessary. To address this issue, we carried out additional experiments where a random permutation layer was inserted between the affine
> > > coupling layers of Real NVP, such that the coordinate correspondence would be eliminated. The empirical results show that the introduction of such random permutations makes no significant impact in terms of accuracy of imputed values. However,
> > > we also noticed that the convergence speed becomes slower possibly due to the fact that the learnable correlation becomes more complex.
> > >
> > > [1] Li, Shen, Bryan Hooi, and Gim Hee Lee. &quot;Identifying through flows for recovering latent
> > > representations.&quot; arXiv preprint arXiv:1909.12555 (2019).
> > >
> > > [2] Kirichenko, Polina, Pavel Izmailov, and Andrew Gordon Wilson. &quot;Why normalizing flows fail to detect
> > > out-of-distribution data.&quot; arXiv preprint arXiv:2006.0854

---

> > > > ### Comment · Reviewer_Twou · 2021-11-28
> > > > **A concern about Assumption 2, the dependencies and the correspondence of ($Xi$,$X_j$) and ($S_i$, $S_j$), and applying MAR to the latent space.**
> > > >
> > > > I would thank the authors for the response and for providing the other references. However, here is my concern:
> > > >
> > > > **(I)**
> > > > I knew the difference between ICA and NF. A further elaboration on the identifiability of ICA is to take us on the same page, which means that one should properly utilize the latent space, be careful about the correspondence between the latent space and observation space, and notice what information/conclusions/assumptions can be used/guaranteed not. (_In short, [1] uses the auxiliary variable to have the identifiability which is not the case of the work._ )
> > > > Unfortunately, Assumption 2 is not the case. One should justify it and provide the condition and further discussion for it because it is fundamental and essential for the correctness of the method. For example, what does the method do, and how does it guarantee that Assumption 2 holds.
> > > >
> > > > **(II)** The authors distinguish correspondence from dependencies for the proposed method.
> > > > However, I didn't quite get it. For example, what would be the case that $X_i$ and $S_j$ have no correspondence but the dependencies of $X_i$ and $X_j$ implies the dependencies between $S_i$ and $S_j$.
> > > >
> > > > **(III)** The **correspondence** matters which is used in the method. As shown in Eqn. (6), it uses the correspondence of $X_i$ and $S_j$ (in the paper it is $Z_i$), because $z_i = f_{\psi}^{-1} (x_i)$, I guess if $x_i$ is missing, the method considers $z_i$ as missing as well by using the correspondence. Furthermore, the MAR assumption is applied to $s_m$/$z_m$ and $s_o$/$z_o$ where MAR is an assumption based on $x_o$ and $x_m$. Without the correspondence, how does the MAR hold in the latent space for the corresponding variable $z_o$ and $z_m$?
> > > >
> > > > **(IV)** As for the **dependencies** in the latent space and observation space, which is also assumed in Assumption 2.
> > > > I am not convinced given the figure in [2] such that to believe Assumption 2 holds for the work in general.
> > > > From the empirical result, we could say when it holds, it works. However, it is essential to include the justification, discussion, and guarantee/condition of the assumption, e.g., that why it should hold, and when it holds, as mentioned in **(I)**.

---

> > > > > ### Author Response · Authors · 2021-11-29
> > > > > **Further response**
> > > > >
> > > > > The points made by the reviewer is well taken but given the limited time and space for the paper, we can address all of the concerns in a future work. We have already empirically illustrated the computational efficiency and accuracy of our methods compared to most recent competitive methods in the area, but, clearly, more work is required in this area to elaborate on some of the issues raised by the reviewer.
> > > > >
> > > > > We also want to note that EMFlow shares similarities with the identifiable iFlow proposed in [1]. In iFlow, the base distribution $P_Z(\cdot|\mathbf{u})$ is a factorized exponential family distribution where the auxiliary variable $\mathbf{u}$ decides the natural parameters though MLP. It is also noted that the exponential families have universal approximation capabilities. In EMFlow, the base distribution is a Gaussian $P_Z(\cdot|\boldsymbol{\mu}$, $\boldsymbol{\Sigma})$, where $\boldsymbol{\mu}(\mathbf{m})$ and $\boldsymbol{\Sigma}(\mathbf{m})$ is conditioned on the auxiliary missing mask $\mathbf{m}$ though the EM procedure in Equation (5). Furthermore, iFlow optimizes the marginal likelihood $P_X(\mathbf{x}|\mathbf{u})$, which is what we optimize in Equation (11). The difference is that EMFlow conditions $\boldsymbol{\mu}(\mathbf{m})$ and $\boldsymbol{\Sigma}(\mathbf{m})$ on _batches_ of $\mathbf{m}$, such that the results of iFlow may not be directly applied to EMFlow. But it can be a direction for future work.

---

> > > > > > ### Comment · Reviewer_Twou · 2021-12-01
> > > > > > **Thanks for the response**
> > > > > >
> > > > > > I would thank the authors for the effort in the response to my concerns.
> > > > > > I would say sorry for some of my initial unclear comments, luckily we make it clear in the discussion session and it turns out the discussion converged at some point.

---

### Official Review · Reviewer_L7zK · 2021-11-02

**Correctness:** 3
**Technical Novelty And Significance:** 2
**Empirical Novelty And Significance:** 3
**Recommendation:** 5
**Confidence:** 3

**Main Review:**

The quality of the paper is generally good. The proposed model combines the online EM and flow models to do imputation, which looks to me a quite reasonable design. The model is evaluated using a number of UCI multi-variable datasets and two image datasets (MNIST and CIFAR-10). The performance is generally better than baselines including MCFlow. Because sampling is avoided by using EM, it converges faster than MCFlow.

The presentation of the paper is quite clear. It is organized, well-written, and quite easy to follow. However, my main concern is that the novelty of the paper seems to be limited, especially when compared with MCFlow; it seems to me that the major difference is that this paper replaces the sampling with the EM algorithm. It is a very reasonable extension, yet it has already been widely studied in various domains and applications.

The authors claim in the introduction section that EM can be applied in an interpretable way, which motivates the authors to use EM in this paper, but this point is not further discussed in the paper. I would be very interested to see discussions about how interpretability can be enhanced by this proposed model.

Another minor typo: in the line below Eq. (2), should it be $\subseteq$ instead of $\in$ between $\mathcal{X}^\prime_i$ and $\mathcal{X}$?

**Summary Of The Paper:**

This paper presents a model named EMFlow, which performs data imputation in the latent space using the online EM algorithm together with the normalizing flow models. The normalizing flow models aim to capture the complete data density $p_X$ and the bidirectional mapping between the data space and the latent space, even when the data is only partially observed. The parameters in the latent space are updated using an online EM algorithm. Thanks to the feature-wise mapping, the dependency between features in the data space is carried over to the latent space and hence the imputation can be done. Evaluation using ten UCI datasets, MNIST, and CIFAR-10 datasets show impressive improvement against baseline models and the convergence is faster than MCFlow.

**Summary Of The Review:**

The presentation is clear, the model is theoretically sound, and experiments show impressive improvement against baseline models for most of the dataset and missing rates. However, the novelty is somewhat limited, especially when compared with MCFlow.

---

> ### Author Response · Authors · 2021-11-21
> **Response to reviewer L7zK**
>
> We sincerely appreciate your comments and suggestions to strengthen our paper, and hope our responses help to address your concerns.
>
> 1. __"my main concern is that the novelty of the paper seems to be limited", "the major difference is that this paper replaces the sampling with the EM algorithm"__
>
>     We think sampling is part of both EMFlow and MCFlow. That is, a sample is taken from the base distribution in the latent space and then goes though the transformation layers. The major difference is that MCFlow has an "implicit" sampler in the latent space, which is a standard MLP. This MLP finds samples of latent vectors by maximizing the log-likelihood of imputed vectors in the data space.
>
>     On the other hand, EMFlow has an "explicit" sampler in the latent space: the online EM explicitly replace the missing parts of latent vectors with the conditional means of the base distribution. Such sampling happens completely in the latent space. To achieve this, we assume a Gaussian base distribution with a general covariance matrix in the latent space that will be learned during the optimization. To the best of our knowledge, Gaussian with unstructured covariance had not been used as the base distribution of NFs for applications other than density estimation (e.g., [1], [2]).
>
> 2. __"I would be very interested to see discussions about how interpretability can be enhanced by this proposed model"__
>
>     The interpretability of EMFlow comes from the explicit sampler in the latent space (i.e. online EM). In MCFlow, the MLP in the latent space is a black box and the difference between the input ($\mathbf{z}$) and output ($\widehat{\mathbf{z}}$) latent vectors is unclear. In EMFlow, with a probabilistic model in the latent space, we know exactly how $\mathbf{z}$ is transformed to $\widehat{\mathbf{z}}$, and $\widehat{\mathbf{z}}$ has higher log-likelihood in the latent space than $\mathbf{z}$.
>
>     Such interpretability bring some possibilities of extending EMFlow. For example, there is established work on obtaining the standard errors of the parameters estimated by EM (e.g., [3], [4]). It would be interesting to see how the standard errors based on latent data can be used to estimate the uncertainty of the imputation in the data space by the use of delta-method that can estimate the propagation on uncertainty.
>
> 3. __"Another minor typo"__
>
>     Thanks for pointing it out and it's fixed.
>
>
> References
>
> [1] Laszkiewicz, Mike, Johannes Lederer, and Asja Fischer. "Copula-Based Normalizing Flows." arXiv preprint arXiv:2107.07352 (2021).
>
> [2] Jaini, Priyank, et al. "Tails of lipschitz triangular flows." International Conference on Machine Learning. PMLR, 2020.
>
> [3] Meng, Xiao-Li, and Donald B. Rubin. "Using EM to obtain asymptotic variance-covariance matrices: The SEM algorithm." Journal of the American Statistical Association 86.416 (1991): 899-909
>
> [4] Jamshidian, Mortaza, and Robert I. Jennrich. "Standard errors for EM estimation." Journal of the Royal Statistical Society: Series B (Statistical Methodology) 62.2 (2000): 257-270.

---

> ### Author Response · Authors · 2021-12-05
> **Further questions/concerns?**
>
> We thank the reviewer again for detailed read and constructive comment, and hope that you find our response satisfying. If you have any further questions or concerns, we are willing to provide more explanations and experiments. Meanwhile, if our previous response addresses your concerns, we sincerely hope that you may rearrange the score.
>
> We want to emphasize again the novelty of this work. The latent space of EMFlow operates in a completely different way than MCFlow: the online EM _explicitly_ performs the imputation while _learns_ the parameters of the base distribution during the training. Such design leads to significant improvement on 1) imputation accuracy, 2) computational efficiency and 3) robustness to hyperparameters and initialization compared to MCFlow and other most recent competitive methods.

---

### Official Review · Reviewer_tdhx · 2021-11-02

**Correctness:** 4
**Technical Novelty And Significance:** 3
**Empirical Novelty And Significance:** 3
**Recommendation:** 6
**Confidence:** 4

**Main Review:**

EMFlow builds most closely upon MCFlow, mentioned in the paper. Both models use normalizing flows (NF) to capture the input data distribution, and MCFlow uses a standard MLP to find a latent vector maximizing the log-likelihood of the missing data. EMFlow uses NFs with EM to maximize the probability of the latent vector (under a probabilistic model — a multivariate normal) corresponding to the missing data. EMFlow has strong empirical results compared to the baseline of MCFlow. While the paper does not compare to GAN-based imputation methods, which might outperform simpler architectures, the lack of comparison seems fair given the ease of training EMFlow relative to a GAN-based method. Particularly looking at convergence traces in Figure 2. Overall the text is clear, well-organized, and easy to follow; there are only minor typos.

Main strengths:
The architecture presented here is relatively straightforward, and does not have many hyperparameters, leading to relatively easy training and implementation. The choice of a multivariate gaussian latent space distribution leads easy conditioning and marginalization, and thus lends itself well to the online EM algorithm presented for imputation. The empirical results are strong, and the methodology is appealing for the reasons listed above. While not far from MCFlows, it provides enough of a conceptual and performance improvement to be a significant contribution to that work.

Main weaknesses and areas to address:
The tasks in the empirical section, while not trivial, are not the most difficult examples. MisGAN, for instance, highlights results on CelebA, a more complicated learning task, as well as a higher-dimensional dataset than any listed in the paper. It might strengthen the paper to present results on a higher-dimensional dataset like CelebA or others. Towards this point, it might also be worth evaluating the FID score of imputed images on a dataset like CelebA, rather than just reporting classification accuracy or RMSE.

The dimensionality and general dataset descriptions are missing from the text and should be included. Particularly of importance is to include the dimensionality of the various datasets analyzed to support the claim that EMFlow works well for high-dimensional data.

The nearest neighbor imputation in MNIST and CIFAR-10 to seed EMFLow is quite reasonable. However, this should be compared to as a baseline as well. It is not clear the difference this warm-start makes in practice. Furthermore, EMFlow is able to be seeded with NN imputation, whereas MCFlow is not. EMFlow produces much better looking and smoother images than MCFlow, which contains far more noise, on CIFAR-10, and these differences should be due to algorithm changes in EMFlow and not NN imputation. The concrete suggestion here is to either report NN imputation as a baseline method on it’s own, or seed EMFlow with a random imputation as a form of ablation experiment. A third type of ablation experiment different from the two mentioned can be left up to the authors. Finally, to what degree is the fast convergence a result of the warm-start imputation?

High-dimensional image data are thought to lie on lower-dimensional manifolds. It is unlikely that the covariance matrix ($\Sigma$) learned in Z-space has full rank. As the data dimensionality continues to scale, it seems this methodology of estimating the empirical covariance will likely not scale as well, even with the robust estimator in (31). Could EMFlow be adapted to work with low-rank covariance estimators? Or perhaps the authors could show further analysis of convergence of the covariance matrix estimation during optimization.

Can EMFLow be adapted to impute categorical data?

**Summary Of The Paper:**

This paper presents a novel imputation method for high-dimensional datasets that typically serve as benchmarks for machine learning methods. This method (EMFlow) innovates by training a normalizing flow network to map input data samples to a multivariate Gaussian, where imputation is performed via an online version of expectation-maximization (EM), which is commonly used for missing data imputation. EMFlow is applied across regression tasks in datasets in the UCI machine learning dataset repository, and standard image classification datasets, MNIST and CIFAR-10. Empirical results show strong performance for missing data imputation, as well as downstream classification from these imputations, and the model design choices and well-constructed architecture make for easy training and fast optimization convergence.

**Summary Of The Review:**

Imputation is a key and general ML problem. EMFlow is an intuitive framework methodologically and in practice seems easy to train and performs well on real data. It is principled in its approach — relying on the rigorous EM algorithm as its base. However, EMFlow (albeit fairly due to training complexity) does not compare to what might be state-of-the-art GAN-based imputation methods. It is also unclear how EMFlow would scale to even higher-dimensional datasets given the modeling assumptions.

---

> ### Author Response · Authors · 2021-11-21
> **Response to reviewer tdhx (part I)**
>
> Thank you for appreciating the strengths of our work and the constructive suggestions for potential improvement. Please find point-wise responses to your concerns below. We also amended our Appendix in the updated manuscript based on your comments.
>
> 1. __"the paper does not compare to GAN-based imputation methods, which might outperform simpler architectures"__
>
>     We did compare EMFlow with GAN-based methods such as MisGAN and GAIN, and spent a fair amount of time for tuning them. For UCI datasets and smaller image datasets (compared to CelebA), EMFlow outperforms MisGAN and GAIN as shown in Table 1 and Table 2 in the paper.
>
> 2. __"The tasks in the empirical section, while not trivial, are not the most difficult examples"__
>
>     The main barrier of applying EMFlow to larger tasks like CelebA is the estimation of covariance matrix in the latent space (as you also mentioned). To have a well-conditioned covariance matrix, a batch size larger than the data dimension is desired, which is impractical for some tasks like CelebA even with the help of super batch introduced in Appendix D.
>
>     We tried some methods for high dimensional covariance estimation ([1] and [2]) but didn’t get very promising results. We admit that it is the current limitation of EMFlow and making EMFlow scale to larger tasks should be the priority of future works.
>
>     But we also want to emphasize that, EMFlow performs very well on datasets of moderate sizes in terms of accuracy and the ease of training (fast convergence, robust to the choice of hype-parameters). EMFlow is also robust to initial imputation (please see our response (5) for further details).
>
> 3. __"The dimensionality and general dataset descriptions are missing from the text and should be included"__
>
>     The description of UCI datasets was already included in Appendix E, and the dimensions of MNIST and CIFAR-10 are provided in the text of Sec 4.2.
>
> 4. __"EMFlow is able to be seeded with NN imputation, whereas MCFlow is not"__
>
>    All the methods experimented in this paper (e.g., EMFlow and MCFlow) use the NN imputation as the starting point on image datasets. So, the use of initial NN imputation is perhaps not a significant contributing factor to the final performance difference between EMFlow and MCFlow.
>
> 5. __"Finally, to what degree is the fast convergence a result of the warm-start imputation?"__
>
>     We added appendix G.3 to investigate the impact of warm-start on the convergence speed. Specifically, we compare the NN imputation and zero imputation as the starting point of imputation on MNIST. We find that the convergence of training loss, test set RMSE and the covariance matrix estimation under both initiation schemes remain similar and relatively insensitive, the initial differences being rather negligible.

---

> > ### Author Response · Authors · 2021-11-21
> > **Response to reviewer tdhx (part II)**
> >
> > 6. __"Could EMFlow be adapted to work with low-rank covariance estimators?"__
> >
> >     Yes, we think low-rank covariance estimator is a promising direction to make EMFlow scale to larger datasets. Instead of assuming a full-rank covariance matrix in the latent space, a low-rank/structured/sparse covariance estimator is more appealing for image datasets. We could possibly adapt some of the earlier works in this area as illustrated by [3] and [4]. But to make such adaption feasible, we expect notable change in the part of EM imputation as NF may no longer be strictly identifiable.
> >
> > 7. __"Or perhaps the authors could show further analysis of convergence of the covariance matrix estimation during optimization?"__
> >
> >      We show the convergence of covariance matrix estimation in terms of Frobenius norm in Appendix G.3.
> >
> > 8. __"Can EMFlow be adapted to impute categorical data?"__
> >
> >     It is another interesting direction of the future work of this paper. One possibility is to replace the RealNVP used in the paper with NFs that directly work on categorical features (e.g. [5], [6], [7]), and adjust the objectives accordingly.
> >
> > References
> >
> > [1] Ledoit, Olivier, and Michael Wolf. "A well-conditioned estimator for large-dimensional covariance matrices." Journal of multivariate analysis 88.2 (2004): 365-411.
> >
> > [2] Won, Joong‐Ho, et al. "Condition‐number‐regularized covariance estimation." Journal of the Royal Statistical Society: Series B (Statistical Methodology) 75.3 (2013): 427-450.
> >
> > [3] Zhou, Sheng-Long, et al. "Sparse and low-rank covariance matrix estimation." Journal of the Operations Research Society of China 3.2 (2015): 231-250.
> >
> > [4] Chen, Xixian, Michael R. Lyu, and Irwin King. "Toward efficient and accurate covariance matrix estimation on compressed data." International Conference on Machine Learning. PMLR, 2017.
> >
> > [5] Hoogeboom, Emiel, et al. "Integer discrete flows and lossless compression." arXiv preprint arXiv:1905.07376 (2019).
> >
> > [6] Tran, Dustin, et al. "Discrete flows: Invertible generative models of discrete data." Advances in Neural Information Processing Systems 32 (2019): 14719-14728.
> >
> > [7] Lippe, Phillip, and Efstratios Gavves. "Categorical normalizing flows via continuous transformations." arXiv preprint arXiv:2006.09790 (2020).

---

> > > ### Comment · Reviewer_tdhx · 2021-12-02
> > > **Response to author follow up**
> > >
> > > Thank you for the thorough discussion and for additional experiments, particularly clarifying the initialization across methods (both using NN across methods, and random initialization optimization). This paper seems broadly applicable and of interest to the community, there are slight limitations in scalability to larger datasets but overall the method seems well-justified. My score remains positive.

---

### Decision · Program_Chairs · 2022-01-20

**Decision:**

Reject

**Comment:**

This paper proposes a data imputation method for MCAR and MAR data by combining EM and normalizing flows.  The paper is clearly written.  The idea is interesting and they show better performance compared to MCFlow and competing methods on ten multivariate UCI data, MNIST and CFAR10 image data.

Issues regarding limited novelty compared to MCFlow was raised.
Issues regarding the validity of Assumption 2 on the dependencies in the latent space and observation space was also raised.